# Linear Pseudo-Measurements Filtering for Tracking a Moving Underwater Target by Observations with Random Delays

**DOI:** 10.3390/s25123757

**Published:** 2025-06-16

**Authors:** Alexey Bosov

**Affiliations:** Federal Research Center “Computer Science and Control” of the Russian Academy of Sciences, 44/2 Vavilova Str., 119333 Moscow, Russia; abosov@frccsc.ru

**Keywords:** autonomous underwater vehicle, stochastic system with randomly delayed observations, linear pseudo-measurements, extended Kalman filter, target tracking, sonars

## Abstract

The linear pseudo-measurements filter is adapted for use in a stochastic observation system with random time delays between the arrival of observations and the actual state of a moving object. The observation model is characterized by limited prior knowledge of the measurement errors distribution, specified only by its first two moments. Furthermore, the proposed model allows for a multiplicative dependence of errors on the state of the moving object. The filter incorporates direction angles and range measurements generated by several independent measurement complexes. As a practical application, the method is used for tracking an autonomous underwater vehicle moving toward a stationary target. The vehicle’s velocity is influenced by continuous random disturbances and periodic abrupt changes. Observations are performed by two stationary acoustic beacons.

## 1. Introduction

Autonomous mobile systems have become a subject of intensive research due to the wide range of fundamental and applied challenges they present. A significant place among these is occupied by the problems of determining the position of a moving object based on indirect observations—state filtering problems in stochastic dynamic systems. Applications of filtering methods and algorithms include, for example, conventional unmanned aerial vehicles [1] and unmanned cars [2]. Autonomous underwater vehicles (AUVs) [3] also belong to this category. This area has recently become a major focus of research. Several factors contribute to this. The aquatic environment poses unique challenges compared to surface navigation, which are created by various factors such as varying temperature, salinity, and water pressure, as well as currents [4,5]. In addition to differences in movement conditions, water also affects the operation of measurement tools. Most of the available measurement tools that are not mounted onboard the AUV use acoustic signals, i.e., they are acoustic sensors or sonars [6]. A key feature of such devices is the significant influence of random time delays in the arrival of data about the observed AUV’s state on measurement accuracy. If the observed vehicle is detected at a distance of several kilometers from the observer, then at the acoustic wave propagation speed of 1500 m/s in water, coordinate data will arrive with a delay of several seconds. The resulting position error may reach several tens of meters and must therefore be taken into account. This issue is addressed by a model of a stochastic observation system with random time delays in measurements. This model is presented in [7,8], and extended in [9,10] to include identification of unknown parameters of the motion model, and the relations of optimal Bayesian filtering are formulated in [11]. For practical application, various suboptimal filtering methods may be used, ranging from the extended Kalman filter (EKF) [12], particle filters [13], and different variants of sigma-point filters [14], up to conditionally optimal and minimax filters [15,16]. The use of optimal Bayesian procedures is not considered due to their excessive computational cost. However, universal methods often perform poorly in systems with delays, with suboptimal algorithms exhibiting a marked tendency to diverge [9].

Similar problems arise when attempting to adapt other, more sophisticated methods to time delays—those aimed at improvement, bias reduction, adaptation to unknown parameters, robustness, etc. Excellent universal methods, from early studies aimed at improving the Kalman filter through repeated iterations [17], improved iterative procedures [18], and the generalized iterative extended Kalman filter for state estimation based on observations with correlated measurement errors [19], struggle to perform under random time delays unless fundamentally modified.

An exception may be the method of linear pseudo-measurements filtering, which occupies an intermediate position between universal methods applicable to any model and specialized ones intended exclusively for a specific model. Although the idea behind this method is fairly universal, it should be applied to specific measurements by linearizing them. In underwater navigation tasks, despite the diversity of sensors used, the measurements are quite uniform—these are direction angles and distances [20].

The pseudo-measurement concept has been known for a long time and represents a logical extension of the most popular and simple suboptimal filtering method, the EKF [12]. The extended filter reproduces the structure of the linear Kalman filter [21], which is optimal for state filtering in a linear-Gaussian observation system. Moreover, the classical linear filter possesses a number of outstanding properties in various robust and adaptive estimation and control problems, which benefits the EKF. Formal adherence to the structure of the linear filter implies linearization. In the case of the EKF, this means linearization around the state prediction to obtain heuristic estimates of the prediction and filtering errors covariances. Accordingly, the quality of the EKF estimate depends significantly on the success of this linearization. A positive result is far from guaranteed, while a negative one may not just be poor. Many works on EKF improvement mention that the filter may diverge. However, this behavior may occur with any other suboptimal filter as well. Systematic studies of this phenomenon are scarce. The authors of [22] considered a simple planar motion model and the possibility of counteracting divergence. The study [23] is also devoted to addressing divergence in a typical robotic control task—simultaneous localization and mapping. A significant theoretical contribution was made in [24] with respect to Bayesian parameter identification, although it did not lead to further development. Naturally, researchers continue to encounter unacceptable EKF behavior [25,26], and this is likely to continue.

Nevertheless, reproducing the structure of the linear Kalman filter remains an attractive idea, especially since we can approach this reproduction from another perspective: instead of tuning the filter, we transform the observation system so that it becomes “similar” to a linear one. This is the core idea of linear pseudo-measurements. From such a transformed system, it is reasonable to expect improved performance of Kalman-type filters. At the same time, it is evident that the feasibility of such a transformation depends on the specific form of the observer function.

The possibility of forming linear pseudo-measurements was apparently first demonstrated for direction angle measurements in [27], although tracking system models based on bearing-only observations began attracting attention even earlier [28]. Initially, this idea did not receive deserved attention. Most likely, the reason was that the actual effect could only be assessed through diverse and large-scale computational experiments, realistic models, and practical applications. At that time, naturally, there were limitations. For example, the simulation in [27] was performed for a planar motion model with a single scalar observation and repeated 30 times. The few studies conducted primarily focused on the issue of filtering estimate bias—a typical deficiency of suboptimal filters [29,30,31,32].

The idea was revisited at a more modern level, compared with other filters, and applied to practical problems in [33]. However, no fundamental changes were introduced: the motion remained planar, and the tangent of the scalar angle measurement was used for the transformation. A new quality was brought to the pseudo-measurement filter by a series of works initiated in [34]. The authors proposed replacing angle measurements with measurements of the tangent, enabling linear pseudo-measurements even in a model where the observer measures two angles—azimuth (bearing) and elevation—and range, with both radio-electronic and acoustic sensors simulated [35,36,37,38], as well as visual odometry data [39,40].

The objective of this paper is to adapt the linear pseudo-measurement method for a model with random time delays and under conditions of incomplete information about the error distribution of measurements. For this purpose, Section 2 proposes a pseudo-measurement model that combines the classical method [27] with a minimax approach. In Section 3, this model is used to derive filtering equations based on the EKF for a stochastic observation system with time delays. Section 4 of the paper is devoted to a computational experiment describing the tracking of AUV motion toward a target, observed by two stationary acoustic beacons. The motion model is complicated by abrupt changes in speed parameters, and the observation model includes measurement error dependence on the object’s range.

## 2. Linear Pseudo-Measurement Filtering (For AUVs and Not Only)

### 2.1. Existing Models

We use the following notations: $E\left\{X\right\}$ denotes the expected value of a random vector X; $cov\left(X,Y\right)$ denotes the covariance X and Y; $X^{\prime }$ denotes the transpose of X. Throughout the remainder of the text, lowercase symbols y will be used to denote actual observations and measurements, while uppercase symbols Y will refer to pseudo-measurement.

We emphasize that at the moment the only application for the model and method discussed below is AUV navigation. Therefore, the terminology used throughout refers to AUV motion and sonar-based measurements. Let us assume that at the moment of detection the underwater vehicle (hereinafter denoted as A) has a coordinate $\left(X_{A},Y_{A},Z_{A}\right)$ in the coordinate system Oxyz (Figure 1), where the plane Oxy coincides with the sea surface, and the axis Oz is directed downward and corresponds to depth. As time t changes during motion, the position of the AUV becomes $\left(X\left(t\right),Y\left(t\right),Z\left(t\right)\right)$.

The measuring device (hereinafter denoted as M, meter), assumed for simplicity to be the only one, is stationary and has coordinates $\left(X_{M},Y_{M},Z_{M}\right)$. The device may be a passive acoustic sensor that estimates the direction of arrival (DOA) for onboard AUV use [41], or an active hydroacoustic beacon that provides measurements to an external observer [42]. The type of measuring device depends on the specific navigation task. If the AUV interacts with the meter (a cooperative scenario), and tracks its own position, a positioning task is solved onboard the AUV. If opposing interests are involved, then an external system solves the task of tracking an unknown moving target. In any case, the measurement of the azimuthal angle φ in the Oxy plane toward the source of the acoustic signal (bearing) is performed with some error:$$y_{\varphi }=\varphi +v_{\varphi },$$
where $v_{\varphi }$ is normally distributed with zero mean and standard deviation $\sigma_{\varphi }$. The method of linear pseudo-measurements, as applied to this model, consists of computing $tan\left(y_{\varphi }\right)$, representing φ as $\varphi =arctan\left(\frac{{Y_{A}-Y}_{M}}{{X_{A}-X}_{M}}\right)$, and transforming $tan\left(\varphi +v_{\varphi }\right)$ using trigonometric identities for the sine and cosine of the sum of angles. As a result, the identity $tan\left(\varphi \right)=\frac{{Y_{A}-Y}_{M}}{{X_{A}-X}_{M}}$ (Figure 1) makes it possible to express a linear combination of the unknown coordinates $\left(X_{A},Y_{A}\right)$ [27,33]. The covariance of the measurement error, which must be provided to the filter, is described by an empirical function involving the unknown state $\left(X_{A},Y_{A}\right)$, which is replaced in the filter by the current estimates.

Another previously mentioned model [34,35,36,37,38,39,40] assumes that the sensors directly measure the tangent (and, consequently, other trigonometric functions of the azimuth), i.e., instead of the previous angle measurement, $y_{\varphi }$ takes a different form:$$y_{\varphi }=\frac{Y_{A}-Y_{M}}{{X_{A}-X}_{M}}+v_{\varphi }.$$

Thus, the observation $y_{\varphi }$ is directly transformed into a linear combination of the measured state $\left(X_{A},Y_{A}\right)$. The advantage of this model lies in the fact that no heuristic assumptions are required for calculating the covariance of the measurement error; the expression for it is obtained without additional assumptions.

The same approach can be applied to the model measuring the tangent of the elevation angle λ (Figure 1):$$y_{\lambda }=\frac{{Z_{A}-Z}_{M}}{\left|X_{A}-X_{M}\right|}cos\left(\varphi \right)+v_{\lambda }.$$

If $y_{\lambda }$ is interpreted as the tangent of the “measured” elevation angle $\tilde{\lambda }$, namely, $y_{\lambda }=\frac{sin\left(\tilde{\lambda }\right)}{cos\left(\tilde{\lambda }\right)}$, and $cos\left(\varphi \right)$ is replaced by $cos\left(arctan\left(y_{\varphi }\right)\right)$, then it is also possible to form a linear combination $\left(X_{A},Y_{A},Z_{A}\right)$.

Finally, a similar approach can be applied to range measurement $r=\frac{Z_{A}-Z_{M}}{sin\left(\lambda \right)}$ (Figure 1). While the specifics of acoustic sensors make angular measurements more relevant for navigation and positioning tasks, in a more familiar aerial environment, range measurement is typical when observing an aerial vehicle [43]. Since there are no fundamental limitations on range measurement—including those based on the Doppler effect or more original techniques based on computer vision [38]—we will further assume the possibility of range measurement. The corresponding model has the following form:$$y_{r}=\frac{Z_{A}-Z_{M}}{sin\left(\lambda \right)}+v_{r},$$
so, to form linear pseudo-measurements, it is sufficient to replace $sin\left(\lambda \right)$ with $sin\left(\tilde{\lambda }\right)$. The same assumptions are made regarding $v_{\lambda }$ and $v_{r}$ as for $v_{\varphi }$, i.e., normal distribution with zero mean and known standard deviations $\sigma_{\lambda }$ and $\sigma_{r}$, respectively.

Note that, depending on the relative positions of the sensor and the AUV, the possible values are φ∈[0,2π], λ∈[0,π] (Figure 1). Determining the direction to the vehicle (clockwise or counterclockwise) from the measurement of $tan\left(\varphi \right)$ is made possible by the sign of $tan\left(\lambda \right)$, due to the observer’s attachment to the water surface, which implies $Z_{A}>Z_{M}$. Hence, in particular, $y_{r}$ must take positive values if the vehicle remains underwater. This, in turn, raises the question of the adequacy of the assumption of normality (or independence) of $v_{r}$.

The method of linear pseudo-measurements is applied for state $X_{t}$ estimation of a discrete stochastic system given observations $y_{t}\in R^{q_{y}}$. Without loss of generality, we assume that the state $X_{t}$ describes the position of the AUV in Cartesian coordinates according to Figure 1, i.e., $X_{t}={\left(X\left(t\right),Y\left(t\right),Z\left(t\right)\right)}^{\prime }$. The estimation of $X_{t}$ starts at time t=0 and is performed at discrete moments 1,2,…,t,…, corresponding to the partition of the observation interval with step δ seconds: δ,2δ,…,tδ,… The AUV’s initial position is defined by the vector $X_{0}=\eta ={\left(\eta_{X},\eta_{Y},\eta_{Z}\right)}^{\prime }={\left(X\left(0\right),Y\left(0\right),Z\left(0\right)\right)}^{\prime }$. $X_{t}$ and $y_{t}$ are described by a discrete stochastic dynamic system of the general form:(1)$$\begin{array}{c} X_{t}=\Phi_{t}^{\left(1\right)}\left(X_{t-1}\right)+\Phi_{t}^{\left(2\right)}\left(X_{t-1}\right)W_{t},t=1,2,\ldots ,X_{0}=\eta ,\\ y_{t}=\psi_{t}^{\left(1\right)}\left(X_{t}\right)+\psi_{t}^{\left(2\right)}\left(X_{t}\right)v_{t}. \end{array}$$

In what follows, it is assumed that the random sequences $X_{t}$ and $y_{t}$ have finite covariances (their existence is ensured, for example, by linear growth constraints on the functions $\Phi_{t}^{\left(1\right)}$, $\Phi_{t}^{\left(2\right)},$ $\psi_{t}^{\left(1\right)}$ and $\psi_{t}^{\left(2\right)}$ [7,8]); the disturbances $W_{t}\in R^{p_{W}}$ and measurement errors $v_{t}\in R^{q_{v}}$ are mutually independent second-order discrete white noise processes; the initial condition vector $\eta \in R^{p_{X}}$ is independent of $W_{t}$ and $v_{t}$ and has a finite covariance. The corresponding first- and second-order moments (mean and covariance of a vector) are denoted (using $W_{t}$ as an example) as $m_{W}\left(t\right),D_{W}\left(t\right).$

In all works, instead of the actual observations $y_{t}$, pseudo-measurements $Y_{t}\in R^{q_{Y}}$ are used in system (1) (since in all existing examples exactly one pseudo-measurement is formed per actual observation, it holds that $q_{Y}=q_{y}$, although in the general case the dimensions may differ):(2)$$Y_{t}=\Psi_{t}^{\left(1\right)}\left(X_{t},y_{t}\right)+\Psi_{t}^{\left(2\right)}\left(X_{t},y_{t}\right)V_{t}$$
and the EKF is applied and used to get an estimate ${\hat{X}}_{t}$. In the adopted notation, it can be written in the following form:(3)$$\begin{array}{c} {\tilde{X}}_{t}=\Phi_{t}^{\left(1\right)}({\hat{X}}_{t-1})+\Phi_{t}^{\left(2\right)}\left({\hat{X}}_{t-1}\right)m_{W}\left(t\right),\\ \begin{array}{c} {\tilde{K}}_{t}={\tilde{\Phi }}_{t}^{\left(1\right)}{\hat{K}}_{t-1}{\left({\tilde{\Phi }}_{t}^{\left(1\right)}\right)}^{\prime }+{\tilde{\Phi }}_{t}^{\left(2\right)}D_{W}\left(t\right){\left({\tilde{\Phi }}_{t}^{\left(2\right)}\right)}^{\prime },\\ {\tilde{\Phi }}_{t}^{\left(1\right)}={\left.\frac{\partial \Phi_{t}^{\left(1\right)}(X)}{\partial X}\right|}_{X={\tilde{X}}_{t}},{\tilde{\Phi }}_{t}^{\left(2\right)}=\Phi_{t}^{\left(2\right)}\left({\tilde{X}}_{t}\right), \end{array}\\ \begin{array}{c} {\hat{X}}_{t}={\tilde{X}}_{t}+K_{t}\left(Y_{t}-\Psi_{t}^{\left(1\right)}({\tilde{X}}_{t},y_{t})-\Psi_{t}^{\left(2\right)}({\tilde{X}}_{t},y_{t})m_{V}\left(t\right)\right),\\ \begin{array}{c} K_{t}={\tilde{K}}_{t}{\left({\tilde{\Psi }}_{t}^{\left(1\right)}\right)}^{\prime }{\left({\tilde{\Psi }}_{t}^{\left(1\right)}{\tilde{K}}_{t}{\left({\tilde{\Psi }}_{t}^{\left(1\right)}\right)}^{\prime }+{\tilde{\Psi }}_{t}^{\left(2\right)}D_{V}\left(t\right){\left({\tilde{\Psi }}_{t}^{\left(2\right)}\right)}^{\prime }\right)}^{-1},\\ \begin{array}{c} {\tilde{\Psi }}_{t}^{\left(1\right)}={\left.\frac{\partial \Psi_{t}^{\left(1\right)}(X,y_{t})}{\partial X}\right|}_{X={\tilde{X}}_{t}},{\tilde{\Psi }}_{t}^{\left(2\right)}=\Psi_{t}^{\left(2\right)}\left({\tilde{X}}_{t},y_{t}\right),\\ {\hat{K}}_{t}={\tilde{K}}_{t}-K_{t}{\tilde{\Psi }}_{t}^{\left(1\right)}{\tilde{K}}_{t}. \end{array} \end{array} \end{array} \end{array}$$

Any of the linear pseudo-measurement variants simplifies the standard expression (3) due to the linearity of $\Psi_{t}^{\left(1\right)}$ with respect to the estimated variables: $\Psi_{t}^{\left(1\right)}\left(X_{t},y_{t}\right)=\Psi_{t}^{\left(1\right)}\left(y_{t}\right)X_{t}$, so that $\frac{\partial \Psi_{t}^{\left(1\right)}(X,y_{t})}{\partial X}={\tilde{\Psi }}_{t}^{\left(1\right)}=\Psi_{t}^{\left(1\right)}\left(y_{t}\right)$. Everything else remains as in the standard EKF: the prediction ${\tilde{X}}_{t}$ due to the system dynamic, the heuristic covariance of the prediction error ${\tilde{K}}_{t}$—the result of linearizing the state equation around the prediction, the correction—the residual of the observations and the Kalman gain $K_{t}$, and the heuristic covariance of the estimation error ${\hat{K}}_{t}$—the result of linearizing the observation equation.

It is usually left uncommented that, unlike the original EKF model, in the pseudo-measurements method the matrices ${\tilde{\Psi }}_{t}^{\left(1\right)}$ and ${\tilde{\Psi }}_{t}^{\left(2\right)}$ in (3) depend not only on the state $X_{t}$ prediction ${\tilde{X}}_{t}$, but also on the actual observations $y_{t}$. For linear pseudo-measurements, the dependence on ${\tilde{X}}_{t}$ is absent precisely because of the linearity, but, instead, a dependence of ${\tilde{\Psi }}_{t}^{\left(1\right)}$ and ${\tilde{\Psi }}_{t}^{\left(2\right)}$ on the measurements $y_{\varphi }$, $y_{\lambda }$, $y_{r}$ arises. Clearly, this can be technically explained easily: it is sufficient to extend the state vector with these variables, write the filter, and then exclude the added variables from the estimate. However, what is more important than these considerations is to note the fundamentally different meaning of the Kalman gain matrix, more precisely, the matrix ${\hat{K}}_{t}$. Specifically, in the traditional EKF, this matrix represents a heuristic estimate of the covariance of the estimation error $cov\left(X_{t}-{\hat{X}}_{t},X_{t}-{\hat{X}}_{t}\right)$. If system (1) is made linear, the filter (3) becomes a linear Kalman filter, i.e., the optimal linear estimate. In this case, the heuristic value ${\hat{K}}_{t}$ becomes the exact covariance of the estimating error. In the pseudo-measurements model, ${\hat{K}}_{t}$ depends on the “actual” observations and is computed at each trajectory, so it should be interpreted as an estimate of the conditional covariance $cov\left(\left.X_{t}-{\hat{X}}_{t},X_{t}-{\hat{X}}_{t}\right|y_{s},0\leq s\leq t\right)$. If we assume that system (1) describes linear dynamics and the observation equations are exact (not approximate as in the pseudo-measurements method), then filter (3) becomes a conditional Gaussian Liptser–Shiryaev filter [44]. In this case, the heuristic estimate of the conditional covariance becomes the exact conditional covariance, characterizing the filtering accuracy on a specific trajectory.

It should be noted that these considerations are not formal and are made in order to mention the most important primary sources in the field of nonlinear filtering. The EKF used here, both traditional and based on the pseudo-measurements method, remains a suboptimal filter that does not have guaranteed statistical characteristics.

### 2.2. Pseudo-Measurements for Unknown Errors Distribution

The proposed approach to constructing linear pseudo-measurements from the observations of φ, λ and r utilizes the idea of the classical filter [27], the tangent observation model [34], as well as the minimax properties of the normal distribution [45].

We first consider the measurement $y_{\varphi }=\varphi +v_{\varphi }$ of the bearing φ with an error $v_{\varphi }$. The distribution of $v_{\varphi }$ is unknown, but it has zero mean $E\left\{v_{\varphi }\right\}=0$ and known mean deviation $\sigma_{\varphi }$: $cov\left(v_{\varphi },v_{\varphi }\right)=\sigma_{\varphi }^{2}$. Assuming that the angular measurement errors of sonars are on the order of ${1^{{}^{\circ}}-2}^{{}^{\circ}}$, we can set $\sigma_{\varphi }=\frac{\pi }{180}\approx 0.0175\;rad$. Therefore, $\sigma_{\varphi }\ll 1$, and consequently $v_{\varphi }\ll 1$.

Based on the given assumption of the smallness of $v_{\varphi }$, and thereafter of other errors arising in the indirect measurements of the AUV position, we will further use simple algebraic approximations for the trigonometric functions of the measured angles. To understand the influence of these empirical assumptions and the errors they introduce into the observation model, let us recall that our ultimate goal is the estimation of the AUV’s position. Thus, by ultimately analyzing the main target error $X_{t}-{\hat{X}}_{t}$, we will also assess the realism of the empirical assumptions made.

From the measurement $y_{\varphi }$, we construct the sine and cosine and approximate them by the corresponding linear terms of the Taylor expansion for small $v_{\varphi }$:$$\begin{array}{c} y_{\varphi }^{sin}=sin\left(y_{\varphi }\right)=sin\left(\varphi +v_{\varphi }\right)\approx sin\left(\varphi \right)+cos\left(\varphi \right)v_{\varphi },\\ y_{\varphi }^{cos}=cos\left(y_{\varphi }\right)=cos\left(\varphi +v_{\varphi }\right)\approx cos\left(\varphi \right)-sin\left(\varphi \right)v_{\varphi }. \end{array}$$

Next, note that $E\left\{{\left(cos\left(\varphi \right)v_{\varphi }\right)}^{2}\right\}\leq \sigma_{\varphi }^{2}$ and $E\left\{{\left(sin\left(\varphi \right)v_{\varphi }\right)}^{2}\right\}\leq \sigma_{\varphi }^{2}$. Moreover, we assume that φ has a distribution symmetric with respect to zero, and therefore $E\left\{cos\left(\varphi \right)v_{\varphi }\;sin\left(\varphi \right)v_{\varphi }\right\}=\frac{1}{2}\sigma_{\varphi }^{2}E\left\{sin\left(2\varphi \right)\right\}=0$. Given the physical meaning of φ, this assumption is quite realistic. We now refer to the well-known minimax property of the normal distribution, which maximizes the variance within the class of distributions with a known mean and bounded covariance matrix [16,45], and construct the following approximation:$$\begin{array}{c} y_{\varphi }^{sin}\approx sin\left(\varphi \right)+v_{1},\\ y_{\varphi }^{cos}\approx cos\left(\varphi \right)+v_{2}, \end{array}$$
where $v_{1}$ and $v_{2}$ are independent Gaussian random variables with $E\left\{v_{1}\right\}=E\left\{v_{2}\right\}=0$ and $E\left\{v_{1}^{2}\right\}=E\left\{v_{2}^{2}\right\}=\sigma_{\varphi }^{2}$. The errors vector ${\left(v_{1},v_{2}\right)}^{\prime }$ is thereby interpreted as the worst-case scenario.

Proceeding with the derivations, we obtain$$\begin{array}{c} sin\left(\varphi \right)\approx y_{\varphi }^{sin}-v_{1},\\ cos\left(\varphi \right)\approx y_{\varphi }^{cos}-v_{2}, \end{array}\;tan\left(\varphi \right)=\frac{{Y_{A}-Y}_{M}}{{X_{A}-X}_{M}}\approx \frac{y_{\varphi }^{sin}-v_{1}}{y_{\varphi }^{cos}-v_{2}},\;\left({Y_{A}-Y}_{M}\right)y_{\varphi }^{cos}-\left(X_{A}-X_{M}\right)y_{\varphi }^{sin}\approx \left(Y_{A}-Y_{M}\right)v_{2}-\left({X_{A}-X}_{M}\right)v_{1}.$$

If, in the last expression, the exact coordinates $\left(X_{A},Y_{A}\right)$ are replaced with their estimates, the result is the residual of the pseudo-measurements derived from the filtering equations. The pseudo-measurement itself is expressed as$$-Y_{M}y_{\varphi }^{cos}+X_{M}y_{\varphi }^{sin}\approx \left(y_{\varphi }^{sin},-y_{\varphi }^{cos}\right)\left(\begin{array}{c} X_{A}\\ Y_{A} \end{array}\right)+\left(Y_{A}-Y_{M}\right)v_{2}-\left({X_{A}-X}_{M}\right)v_{1},$$
which clarifies the meaning of the performed transformations: the pseudo-measurements $-Y_{M}y_{\varphi }^{cos}+X_{M}y_{\varphi }^{sin}$ approximate the measurement of a linear combination of the estimated coordinates $\left(X_{A},Y_{A}\right)$ in the presence of additive noise with known covariance.

Thus, for the measurement $y_{\varphi }=\varphi +v_{\varphi }$ of the bearing φ, the pseudo-measurement $Y_{\varphi }$ is constructed as(4)$$Y_{\varphi }=-Y_{M}y_{\varphi }^{cos}+X_{M}y_{\varphi }^{sin},y_{\varphi }^{sin}=sin\left(y_{\varphi }\right),y_{\varphi }^{cos}=cos\left(y_{\varphi }\right),$$
and the filtering algorithm employs the observation model of the form(5)$$Y_{\varphi }=\left(y_{\varphi }^{sin},-y_{\varphi }^{cos}\right)\left(\begin{array}{c} X_{A}\\ Y_{A} \end{array}\right)+\left(\begin{array}{cc} X_{M}-X_{A}, & Y_{A}-Y_{M} \end{array}\right)\left(\begin{array}{c} v_{1}\\ v_{2} \end{array}\right).$$

Next, consider the measurement $y_{\lambda }=\lambda +v_{\lambda }$ for the elevation angle λ. Similarly to the bearing, we approximate the sine and cosine, based on the same assumptions about the measurement error $v_{\lambda }$:$$\begin{array}{c} y_{\lambda }^{sin}=sin\left(y_{\lambda }\right)\approx sin\left(\lambda \right)+cos\left(\lambda \right)v_{\lambda }\approx sin\left(\lambda \right)+v_{3},\\ y_{\lambda }^{cos}=cos\left(y_{\lambda }\right)\approx cos\left(\lambda \right)-sin\left(\lambda \right)v_{\lambda }\approx cos\left(\lambda \right)+v_{4} \end{array}$$
with independent Gaussian random variables $v_{3}$ and $v_{4}$, $E\left\{v_{3}\right\}=E\left\{v_{4}\right\}=0$ and $E\left\{v_{3}^{2}\right\}=E\left\{v_{4}^{2}\right\}=\sigma_{\lambda }^{2}$. From this, we get$$\begin{array}{c} sin\left(\lambda \right)\approx y_{\lambda }^{sin}-v_{3},\\ cos\left(\lambda \right)\approx y_{\lambda }^{cos}-v_{4}, \end{array}tan\left(\lambda \right)=\frac{Z_{A}-Z_{M}}{\left|X_{A}-X_{M}\right|}cos\left(\varphi \right)\approx \frac{y_{\lambda }^{sin}-v_{3}}{y_{\lambda }^{cos}-v_{4}}.$$

To simplify the manipulations with the measurement $y_{\lambda }$, we will assume that the relative positioning of A and M is taken into account when choosing the coordinate system, so that $X_{A}>X_{M}$ and $X\left(t\right)>X_{M}$ during motion. Using the existing bearing approximation, we replace $cos\left(\varphi \right)$ with $y_{\varphi }^{cos}-v_{2}$ and obtain$$\begin{array}{c} y_{\varphi }^{cos}y_{\lambda }^{cos}\left(Z_{A}-Z_{M}\right)-y_{\lambda }^{sin}\left(X_{A}-X_{M}\right)\approx \\ \approx \left(Z_{A}-Z_{M}\right)y_{\lambda }^{cos}v_{2}-\left(X_{A}-X_{M}\right)v_{3}+\left(Z_{A}-Z_{M}\right)y_{\varphi }^{cos}v_{4}-\left(Z_{A}-Z_{M}\right)v_{2}v_{4}. \end{array}$$

Since all $v_{i}$ were assumed to be independent and centered, the variance of the right-hand side of the obtained expression (the errors of the pseudo-measurement) is given by$$\begin{array}{c} E\left\{{\left(\left(Z_{A}-Z_{M}\right)y_{\lambda }^{cos}v_{2}-\left(X_{A}-X_{M}\right)v_{3}+\left(Z_{A}-Z_{M}\right)y_{\varphi }^{cos}v_{4}-\left(Z_{A}-Z_{M}\right)v_{2}v_{4}\right)}^{2}\right\}=\\ ={\left(Z_{A}-Z_{M}\right)}^{2}{\left(y_{\lambda }^{cos}\right)}^{2}\sigma_{\varphi }^{2}+{\left(X_{A}-X_{M}\right)}^{2}\sigma_{\lambda }^{2}+{\left(Z_{A}-Z_{M}\right)}^{2}{\left(y_{\varphi }^{cos}\right)}^{2}\sigma_{\lambda }^{2}+{\left(Z_{A}-Z_{M}\right)}^{2}\sigma_{\varphi }^{2}\sigma_{\lambda }^{2}. \end{array}$$

Appealing to the same assumptions about the small values of $v_{i}$ and $\sigma_{\varphi }$, $\sigma_{\lambda }$, we neglect the last term and represent the residual of the pseudo-measurement (when the exact coordinates $\left(X_{A},Y_{A},Z_{A}\right)$ are replaced by their estimates) as$$\begin{array}{c} y_{\varphi }^{cos}y_{\lambda }^{cos}\left(Z_{A}-Z_{M}\right)-y_{\lambda }^{sin}\left(X_{A}-X_{M}\right)\approx \\ \approx \left(Z_{A}-Z_{M}\right)y_{\lambda }^{cos}v_{2}-\left(X_{A}-X_{M}\right)v_{3}+\left(Z_{A}-Z_{M}\right)y_{\varphi }^{cos}v_{4}, \end{array}$$
and the pseudo-measurement itself will be written as$$\begin{array}{c} -y_{\varphi }^{cos}y_{\lambda }^{cos}Z_{M}+y_{\lambda }^{sin}X_{M}\approx \\ \approx -y_{\varphi }^{cos}y_{\lambda }^{cos}Z_{A}+y_{\lambda }^{sin}X_{A}+\left(Z_{A}-Z_{M}\right)y_{\lambda }^{cos}v_{2}-\left(X_{A}-X_{M}\right)v_{3}+\left(Z_{A}-Z_{M}\right)y_{\varphi }^{cos}v_{4}. \end{array}$$

Thus, for the measurement $y_{\lambda }=\lambda +v_{\lambda }$ of the elevation angle λ, the pseudo-measurement $Y_{\lambda }$ is constructed as(6)$$\begin{array}{c} Y_{\lambda }=-y_{\varphi }^{cos}y_{\lambda }^{cos}Z_{M}+y_{\lambda }^{sin}X_{M},\\ \;y_{\lambda }^{sin}=sin\left(y_{\lambda }\right),\;y_{\lambda }^{cos}=cos\left(y_{\lambda }\right),\;y_{\varphi }^{cos}=cos\left(y_{\varphi }\right), \end{array}$$
and the filtering algorithm uses the observation model of the form(7)$$\begin{array}{c} Y_{\lambda }=\left(y_{\lambda }^{sin},-y_{\varphi }^{cos}y_{\lambda }^{cos}\right)\left(\begin{array}{c} X_{A}\\ Z_{A} \end{array}\right)+\\ +\left(\begin{array}{ccc} \left(Z_{A}-Z_{M}\right)y_{\lambda }^{cos}, & X_{M}-X_{A}, & \left(Z_{A}-Z_{M}\right)y_{\varphi }^{cos} \end{array}\right)\left(\begin{array}{c} v_{2}\\ \begin{array}{c} v_{3}\\ v_{4} \end{array} \end{array}\right). \end{array}$$

Lastly, consider the measurement $y_{r}=r+v_{r}$ of the range r with an error $v_{r}$ independent of previous errors, $E\left\{v_{r}\right\}=0$, $E\left\{v_{r}^{2}\right\}=\sigma_{r}^{2}$. Using the measurement of the elevation angle λ and the approximation $sin\left(\lambda \right)\approx y_{\lambda }^{sin}-v_{3},$ we write$$r=\frac{Z_{A}-Z_{M}}{sin\left(\lambda \right)}\Rightarrow y_{r}-v_{r}\approx \frac{Z_{A}-Z_{M}}{y_{\lambda }^{sin}-v_{3}}$$

Proceeding similarly to the transformations of the angles, we obtain$$Z_{A}-Z_{M}-y_{r}y_{\lambda }^{sin}\approx -y_{r}v_{3}-y_{\lambda }^{sin}v_{r}+v_{3}v_{r}$$

The centered error on the right-hand side has variance$$E\left\{{\left(-y_{r}v_{3}-y_{\lambda }^{sin}v_{r}+v_{3}v_{r}\right)}^{2}\right\}=y_{r}^{2}\sigma_{\lambda }^{2}+{\left(y_{\lambda }^{sin}\right)}^{2}\sigma_{r}^{2}+\sigma_{\lambda }^{2}\sigma_{r}^{2}$$

Here, the third term can be neglected compared to the first two, and the residual of the pseudo-measurement (when the exact coordinate $Z_{A}$ is replaced by its estimate) can be written as$$Z_{A}-Z_{M}-y_{r}y_{\lambda }^{sin}\approx -y_{r}v_{3}-y_{\lambda }^{sin}v_{5}$$
with independent Gaussian $v_{5}$, $E\left\{v_{5}\right\}=0$ and $E\left\{v_{5}^{2}\right\}=\sigma_{r}^{2}$.

The pseudo-measurement itself will be written as$$Z_{M}+y_{r}y_{\lambda }^{sin}\approx Z_{A}+y_{r}v_{3}+y_{\lambda }^{sin}v_{5}$$

Thus, for the measurement $y_{r}=d+v_{5}$ of the range, the pseudo-measurement $Y_{r}$ is constructed as(8)$$Y_{r}=Z_{M}+y_{r}y_{\lambda }^{sin},y_{\lambda }^{sin}=sin\left(y_{\lambda }\right),$$
and the filtering algorithm uses the observation model of the form(9)$$Y_{r}=Z_{A}+\left(\begin{array}{cc} y_{r}, & y_{\lambda }^{sin} \end{array}\right)\left(\begin{array}{c} v_{3}\\ v_{5} \end{array}\right).$$

Now, all three models (5), (7), (9) can be combined into a single vector $Y={\left(Y_{\varphi },Y_{\lambda },Y_{r}\right)}^{\prime }$ of the form(10)$$\begin{array}{c} Y=\left(\begin{array}{ccc} y_{\varphi }^{sin} & -y_{\varphi }^{cos} & 0\\ y_{\lambda }^{sin} & 0 & -y_{\varphi }^{cos}y_{\lambda }^{cos}\\ 0 & 0 & 1 \end{array}\right)X+\\ +\left(\begin{array}{cc} \begin{array}{ccc} X_{M}-X_{A} & Y_{A}-Y_{M} & 0\\ 0 & \left(Z_{A}-Z_{M}\right)y_{\lambda }^{cos} & X_{M}-X_{A}\\ 0 & 0 & y_{r} \end{array} & \begin{array}{cc} \begin{array}{c} 0\\ \left(Z_{A}-Z_{M}\right)y_{\varphi }^{cos}\\ 0 \end{array} & \begin{array}{c} 0\\ 0\\ y_{\lambda }^{sin} \end{array} \end{array} \end{array}\;\right)V, \end{array}$$
where $X={\left(X_{A},Y_{A},Z_{A}\right)}^{\prime }$, $V={\left(v_{1},v_{2},v_{3},v_{4},v_{5}\right)}^{\prime }$, and in motion $X={\left(X\left(t\right),Y\left(t\right),Z\left(t\right)\right)}^{\prime }$.

We now assume that the measurements $y_{\varphi },y_{\lambda },y_{r}$ are taken at each time moment t=0,1,…, forming the observation vector $y_{t}$ in (2). Accordingly, expressions (4), (6), (8) define how to compute the values of the pseudo-measurements $Y_{t}$, and Equation (9) shows how the functions appear in the model (2). All the listed components can be substituted into the algorithm (3) to obtain the state $X_{t}$ estimates from (1).

### 2.3. Pseudo-Measurements with Random Time Delay

The dependence of the observations $y_{t}$ and pseudo-measurements $Y_{t}$ on the state $X_{t}$ changes significantly if the time for information exchange between the observed object and the observer cannot be neglected. This is particularly relevant for sonar systems, as previously noted. Accordingly, the measurements are made for the position where A was at some time s<t. This moment is defined as follows.

Let $v_{s}=const$ be the constant speed of sound in water. For the target tracking problem, this simplification is sufficient, although more accurate algorithms for calculating $v_{s}$ may be required in more sensitive areas, such as those in [46,47]. Regardless of the type of sonar and the location of the measurements (on board or by an external complex), there is a difference between the time when the measurement was obtained and the time when A was in the “measured” position. This difference is the time it takes for the acoustic signal to travel the distance between A and M, i.e., $\tau =t-s=r/\left(\delta v_{s}\right)$. In line with the pseudo-measurements approach, this random value can be approximated by $\tilde{\tau }=y_{r}/\left(\delta v_{s}\right)$. Given that the typical value of $v_{s}=5400\;km/h$ (1500 m/s), the error introduced can be neglected, and the final model of pseudo-measurements (10) can be supplemented with the relation(11)$$X={\left(X\left(t-\tilde{\tau }\right),Y\left(t-\tilde{\tau }\right),Z\left(t-\tilde{\tau }\right)\right)}^{\prime },\;\;\;\tilde{\tau }=y_{r}/\delta v_{s}.$$

The general form of the observation–pseudo-measurements system is as follows:(12)$$\begin{array}{c} X_{t}=\Phi_{t}^{\left(1\right)}\left(X_{t-1}\right)+\Phi_{t}^{\left(2\right)}\left(X_{t-1}\right)W_{t},\;t=-T,-T+1,\ldots ,1,2,\ldots ,\;X_{-T-1}=\eta ,\\ \begin{array}{c} y_{t}=\psi_{t}^{\left(1\right)}\left(X_{t-\tau_{t}}\right)+\psi_{t}^{\left(2\right)}\left(X_{t-\tau_{t}}\right)v_{t},\;\;\;\tau_{t}=\tau_{t}\left(X_{t}\right),\\ Y_{t}=\Psi_{t}^{\left(1\right)}\left(y_{t}\right)X_{t-{\tilde{\tau }}_{t}}+\Psi_{t}^{\left(2\right)}\left(X_{t-{\tilde{\tau }}_{t}},y_{t}\right)V_{t},\;\;\;{\tilde{\tau }}_{t}={\tilde{\tau }}_{t}\left(y_{t}\right). \end{array} \end{array}$$

In this model, it is assumed that the value Tδ>0 is known—the maximum possible time delay of the observations (in fact, the maximum detection range of a moving object) and that the motion of A begins at time −Tδ, i.e., t=−T, so that at time t=0, the observer M can reliably perform the measurement. The initial position of A is defined by the vector $\eta ={\left(\eta_{X},\eta_{Y},\eta_{Z}\right)}^{\prime }={\left(X\left(-T-1\right),Y\left(-T-1\right),Z\left(-T-1\right)\right)}^{\prime }$. The time delay $\tau_{t}$ is a function of the state $X_{t}$. It is equal to the time required for the sound wave to travel the distance between A and M (this is why the estimate $\tilde{\tau }$ is included in the pseudo-measurements (11)).

The functions $\tau_{t}\left(X\right)$ and ${\tilde{\tau }}_{t}\left(y\right)$ in (12) must take integer values from the set $\left\{0,\;1,\;\ldots ,T\right\}$. For actual states and observations, they are given by the following:(13)$$\begin{array}{c} \tau_{t}=min\left\{T,\left[\sqrt{{\left(X\left(t\right)-X_{M}\right)}^{2}+{\left(Y\left(t\right)-Y_{M}\right)}^{2}+{\left(Z\left(t\right)-Z_{M}\right)}^{2}}/\left({\delta v}_{s}\right)\right]\right\},\\ {\tilde{\tau }}_{t}=min\left\{T,\left[y_{r}/\left(\delta v_{s}\right)\right]\right\}, \end{array}$$
where $\left[\cdot \right]$ denotes the integer part of the number.

### 2.4. Model with Multiple Observers

Let the observation vector $y_{t}$ in (12) now combine angle and range measurements received from q observers, i.e., $y_{t}={\left(y_{\varphi_{t}}^{\left(1\right)},y_{\lambda_{t}}^{\left(1\right)},y_{r_{t}}^{\left(1\right)},\ldots ,y_{\varphi_{t}}^{\left(q\right)},y_{\lambda_{t}}^{\left(q\right)},y_{r_{t}}^{\left(q\right)}\right)}^{\prime }$. Since $y_{t}\in R^{q_{y}}$, we obtain $q_{y}=3q$. For each i-th observer, define the time delay $\tau_{t}^{\left(i\right)}$, i=1,…,q, with values in the set $\left\{0,1,\ldots ,T\right\}$. The values $\tau_{t}^{\left(i\right)}$ are collected into the vector $\tau_{t}={\left(\tau_{t}^{\left(1\right)},\ldots ,\tau_{t}^{\left(q\right)}\right)}^{\prime }\in R^{q}$, which is a function of $X_{t}$ in the same manner as $\tau_{t}$ in (12). Thus, the measurements in each group $y_{\varphi_{t}}^{\left(i\right)},y_{\lambda_{t}}^{\left(i\right)},y_{r_{t}}^{\left(i\right)}$ can be represented as functions of the position $X_{t-\tau_{t}^{\left(i\right)}}$. As a result, the observation system takes the form:(14)$$\begin{array}{c} X_{t}=\Phi_{t}^{\left(1\right)}\left(X_{t-1}\right)+\Phi_{t}^{\left(2\right)}\left(X_{t-1}\right)W_{t},\;t=-T,-T+1,\ldots ,0,1,\ldots \;X_{-T-1}=\eta ,\\ \begin{array}{c} y_{t}^{\left(i\right)}=\psi_{t}^{\left(i,1\right)}\left(X_{t-\tau_{t}^{\left(i\right)}}\right)+\psi_{t}^{\left(i,2\right)}\left(X_{t-\tau_{t}^{\left(i\right)}}\right)v_{t}^{\left(i\right)},\;i=1,\ldots ,q,\\ Y_{t}^{\left(i\right)}=\Psi_{t}^{\left(i,1\right)}\left(y_{t}^{\left(i\right)}\right)X_{t-{\tilde{\tau }}_{t}^{\left(i\right)}}+\Psi_{t}^{\left(i,2\right)}\left(X_{t-{\tilde{\tau }}_{t}^{\left(i\right)}},y_{t}^{\left(i\right)}\right)V_{t}^{\left(i\right)}. \end{array} \end{array}$$

To ensure that (14) reflects the stated assumptions and reduces to (1), (2) when T=0, the following notations are required:$$\begin{array}{c} y_{t}={\left({\left(y_{t}^{\left(1\right)}\right)}^{\prime },\ldots ,{\left(y_{t}^{\left(q\right)}\right)}^{\prime }\right)}^{\prime }\;\mathrm{is}\;\mathrm{the}\;\mathrm{vector}\;\mathrm{consisting}\;\mathrm{of}\;q\;\mathrm{groups}\;\mathrm{of}\;\mathrm{measurements},\\ \mathrm{where}\;\mathrm{each}\;\mathrm{group}\;\mathrm{is}\;\mathrm{defined}\;\mathrm{as}\;y_{t}^{\left(i\right)}={\left(y_{\varphi_{t}}^{\left(i\right)},y_{\lambda_{t}}^{\left(i\right)},y_{r_{t}}^{\left(i\right)}\right)}^{\prime },\\ v_{t}^{\left(i\right)}\;\mathrm{is}\;\mathrm{the}\;\mathrm{vector}\;\mathrm{of}\;\mathrm{measurement}\;\mathrm{errors}\;\mathrm{in}\;\mathrm{the}\;i\text{-}\mathrm{th}\mathrm{group};\\ Y_{t}={\left({\left(Y_{t}^{\left(1\right)}\right)}^{\prime },\ldots ,{\left(Y_{t}^{\left(q\right)}\right)}^{\prime }\right)}^{\prime }\;\mathrm{is}\;\mathrm{the}\;\mathrm{vector}\;\mathrm{of}\;q\;\mathrm{groups}\;\mathrm{of}\;\mathrm{pseudo-measurements},\\ \mathrm{where}\;Y_{t}^{\left(i\right)}={\left(Y_{t}^{\left(i\right)},Y_{t}^{\left(i\right)},Y_{t}^{\left(i\right)}\right)}^{\prime }\;\mathrm{corresponds}\;\mathrm{to}\;\mathrm{the}\;i\text{-}\mathrm{th}\;\mathrm{group}\;y_{t}^{\left(i\right)},\\ V_{t}^{\left(i\right)}\mathrm{is}\;\mathrm{the}\;\mathrm{vector}\;\mathrm{of}\;\mathrm{pseudo-measurement}\;\mathrm{errors}\;\mathrm{in}\;\mathrm{this}\;\mathrm{group};\\ \mathrm{the}\;\mathrm{vector}\;\mathrm{functions}\;\psi_{t}^{\left(i,1\right)}\;\mathrm{and}\;\mathrm{matrices}\;\psi_{t}^{\left(i,2\right)},\;\Psi_{t}^{\left(i,1\right)},\;\Psi_{t}^{\left(i,2\right)},\;i=1,\ldots ,q,\;\\ \mathrm{are}\;\mathrm{defined}\;\mathrm{for}\;\mathrm{each}\;\mathrm{group}\;\mathrm{based}\;\mathrm{on}\;\mathrm{the}\;\mathrm{assumption}\;\mathrm{of}\;\mathrm{observer}\;\mathrm{independence},i.e.,\\ \psi_{t}^{\left(1\right)}={\left({\left(\psi_{t}^{\left(1,1\right)}\right)}^{\prime },\ldots ,{\left(\psi_{t}^{\left(q,1\right)}\right)}^{\prime }\right)}^{\prime },\;\psi_{t}^{\left(2\right)}=diag\left(\psi_{t}^{\left(1,2\right)},\ldots ,\psi_{t}^{\left(q,2\right)}\right),\\ \Psi_{t}^{\left(1\right)}=\left(\begin{array}{c} \Psi_{t}^{\left(1,1\right)}\\ \ldots \\ \Psi_{t}^{\left(q,1\right)} \end{array}\right),\;\Psi_{t}^{\left(2\right)}=diag\left(\Psi_{t}^{\left(1,2\right)},\ldots ,\Psi_{t}^{\left(q,2\right)}\right). \end{array}$$

The basic equations of the recursive filtering algorithm (3) can now be refined for the model with time delays (14).(15)$$\begin{array}{c} {\tilde{X}}_{t}=\Phi_{t}^{\left(1\right)}\left({\hat{X}}_{t-1}\right)+\Phi_{t}^{\left(2\right)}\left({\hat{X}}_{t-1}\right)m_{W}\left(t\right),\\ \begin{array}{c} {\tilde{K}}_{t}={\tilde{\Phi }}_{t}^{\left(1\right)}{\hat{K}}_{t-1}{\left({\tilde{\Phi }}_{t}^{\left(1\right)}\right)}^{\prime }+{\tilde{\Phi }}_{t}^{\left(2\right)}D_{W}\left(t\right){\left({\tilde{\Phi }}_{t}^{\left(2\right)}\right)}^{\prime },\\ {\tilde{\Phi }}_{t}^{\left(1\right)}={\left.\frac{\partial \Phi_{t}^{\left(1\right)}(X)}{\partial X}\right|}_{X={\tilde{X}}_{t}},\;\;{\tilde{\Phi }}_{t}^{\left(2\right)}=\Phi_{t}^{\left(2\right)}\left({\tilde{X}}_{t}\right), \end{array}\\ \begin{array}{c} \begin{array}{c} {\hat{X}}_{t}={\tilde{X}}_{t}+K_{t}∆{\tilde{Y}}_{t},\;\\ ∆{\tilde{Y}}_{t}=\left(Y_{t}^{\left(1\right)}-\Psi_{t}^{\left(1,1\right)}\left(y_{t}^{\left(1\right)}\right){\tilde{X}}_{t-{\tilde{\tau }}_{t}^{\left(1\right)}},\ldots ,Y_{t}^{\left(q\right)}-\Psi_{t}^{\left(q,1\right)}\left(y_{t}^{\left(q\right)}\right){\tilde{X}}_{t-{\tilde{\tau }}_{t}^{\left(q\right)}}\right),\; \end{array}\\ \begin{array}{c} K_{t}={\tilde{K}}_{t}{\left(\Psi_{t}^{\left(1\right)}\right)}^{\prime }{\left(\Psi_{t}^{\left(1\right)}{\tilde{K}}_{t}{\left(\Psi_{t}^{\left(1\right)}\right)}^{\prime }+{\tilde{\Psi }}_{t}^{\left(2\right)}D_{V}\left(t\right){\left({\tilde{\Psi }}_{t}^{\left(2\right)}\right)}^{\prime }\right)}^{-1},\\ \begin{array}{c} \Psi_{t}^{\left(1\right)}=\Psi_{t}^{\left(1\right)}\left(y_{t}\right),\;{\tilde{\Psi }}_{t}^{\left(2\right)}=diag\left\{\Psi_{t}^{\left(1,2\right)}\left({\tilde{X}}_{t-{\tilde{\tau }}_{t}^{\left(1\right)}},y_{t}\right),\ldots ,\Psi_{t}^{\left(q,2\right)}\left({\tilde{X}}_{t-{\tilde{\tau }}_{t}^{\left(q\right)}},y_{t}\right)\right\},\\ {\hat{K}}_{t}={\tilde{K}}_{t}-K_{t}\Psi_{t}^{\left(1\right)}{\tilde{K}}_{t}. \end{array} \end{array} \end{array} \end{array}$$

In essence, the filter representation (15), in comparison with filter (3), simply incorporates the estimates of the time delays ${\tilde{\tau }}_{t}^{\left(i\right)}$, i=1,…q, for each observer. The pseudo-measurements residual $∆{\tilde{Y}}_{t}$ and the measurement errors deviation matrix $\Psi_{t}^{\left(2\right)}$ are constructed from the predicted position values ${\tilde{X}}_{t-{\tilde{\tau }}_{t}^{\left(q\right)}}$, corresponding to the time moments at which the current observations $y_{t}$ were obtained and the pseudo-measurements $Y_{t}$ were formed. To achieve this, the position predictions are shifted relative to the current time by the estimated time delay ${\tilde{\tau }}_{t}^{\left(i\right)}$ corresponding to the estimated value of $\tau_{t}^{\left(i\right)}$ for the i-th observer.

Concluding this section, we list the steps performed to compute the estimate of algorithm (15).

Step 0. From the beginning of observation acquisition until the specified time boundary T, we accumulate measurements $y_{t}^{\left(i\right)}$, i=1,…q, received from all q observers.

Step 1. Proceed to compute the estimate at the current step t>0 (using the estimate from the previous step ${\hat{X}}_{t-1}$ and all predictions ${\tilde{X}}_{t-\tau }$ with delays τ=1,…,T).

Step 2. Compute the current prediction ${\tilde{X}}_{t}$ (standard EKF procedure).

Step 3. Compute the empirical accuracy ${\tilde{K}}_{t}$ of the current prediction ${\tilde{X}}_{t}$ (standard EKF procedure).

Step 4. Compute the estimates of the time delays ${\tilde{\tau }}_{t}^{\left(i\right)}$, i=1,…q, for all q observers.

Step 5. Form the vector ${\left({\tilde{X}}_{t-{\tilde{\tau }}_{t}^{\left(1\right)}},\ldots ,{\tilde{X}}_{t-{\tilde{\tau }}_{t}^{\left(q\right)}}\right)}^{\prime }$ of delayed predictions, the corresponding vector of pseudo-measurements $Y_{t}$ and the observation matrixes $\Psi_{t}^{\left(1\right)}$ and $\Psi_{t}^{\left(2\right)}$.

Step 6. Compute the Kalman gain $K_{t}$, the pseudo-measurements residuals $∆{\tilde{Y}}_{t}$, and the filtering estimate ${\hat{X}}_{t}$ (standard EKF procedure).

Step 7. Compute the empirical accuracy ${\hat{K}}_{t}$ of the current estimate ${\hat{X}}_{t}$ (standard EKF procedure) and proceed to Step 1.

## 3. Tracking the Approach of an AUV on Two Sets of Measurements

### 3.1. AUV Motion Model

The model used in [9,10] for the analysis of the AUV motion parameter identification algorithm incorporates measurements from two stationary acoustic beacons. For the analysis of algorithm (15) in this paper, we employ a similar model. The modifications concern, firstly, the velocity parameters in the motion model, which are assumed to be known; since our focus is on analyzing the quality of the filtering estimation, we assume these parameters are either known a priori or have been previously identified. Secondly, we introduce greater complexity into the motion model by modifying the external inputs affecting the velocity, which impart a chaotic nature to the motion. Specifically, we assume that the magnitude of the additive disturbance to the velocity depends on the absolute value of the velocity: the higher the speed, the stronger the acting disturbance. Finally, we introduce several complexities into the observation model to better reflect practical conditions. This primarily concerns measurement errors. The traditional assumption of their additivity and independence from the state is not always justifiable. In the considered example, we reject this assumption for range measurements, reasonably assuming that their accuracy depends on the distance between the AUV and the object.

We use Figure 1 as the basis and further state that an object (O,object), which serves as the AUV’s target, is located at the origin O of the Oxyz coordinate system on the surface and remains stationary. The initial position A is defined by the vector $\eta ={\left(\eta_{X},\eta_{Y},\eta_{Z}\right)}^{\prime }$, whose components are independent and uniformly distributed: $\eta_{X}\sim R[10,\;20]$, $\eta_{Y}\sim R[10,\;20]$, $\eta_{Z}\sim R[0.5,1.5]$. The moments of this distribution are $E\left\{\eta \right\}={\left(15,15,1\right)}^{\prime }$ and $cov\left(\eta ,\eta \right)\approx diag\left\{2.9^{2},2.9^{2},0.29^{2}\right\}$. All distances are measured in kilometers.

The AUV, located at $X_{t}={\left(X\left(t\right),Y\left(t\right),Z\left(t\right)\right)}^{\prime }$, moves toward point O with a piecewise-constant mean velocity. To model the mean velocity of the AUV, we use a random sequence $s\left(t\right)={\left(s_{X}\left(t\right),s_{Y}\left(t\right),s_{Z}\left(t\right)\right)}^{\prime }$. Its distribution is assumed to be known and is described below.

The actual velocity $S\left(t\right)={\left(S_{X}\left(t\right),S_{Y}\left(t\right),S_{Z}\left(t\right)\right)}^{\prime }$ deviates from the mean by an amount described by the vector of additive disturbances ${\left(w_{X}\left(t\right),w_{Y}(t),w_{Z}(t)\right)}^{\prime }$. The components of this vector are independent and follow a standard normal distribution. Thus, for the case of motion described by a model with independent disturbances, the state equation takes the following form:(16)$$\begin{array}{c} \begin{array}{c} X\left(t\right)=X\left(t-1\right)+\delta S_{X}\left(t\right),\;\;\;S_{X}\left(t\right)=s_{X}\left(0\right)+\sigma_{s_{X}}w_{X}\left(t\right),\\ Y\left(t\right)=Y\left(t-1\right)+\delta S_{Y}\left(t\right),\;\;\;S_{Y}\left(t\right)=s_{Y}\left(0\right)+\sigma_{s_{Y}}w_{Y}\left(t\right), \end{array}\\ Z\left(t\right)=Z\left(t-1\right)+\delta S_{Z}\left(t\right),\;\;\;S_{Z}\left(t\right)=s_{Z}\left(0\right)+\sigma_{s_{Z}}w_{Z}\left(t\right). \end{array}$$

The values $\sigma_{s_{X}}$, $\sigma_{s_{Y}}$ and $\sigma_{s_{Z}}$ are known and define the standard deviations of the additive velocity disturbances; δ is the discretization interval defined above for model (1).

In (16), only the initial value $s\left(0\right)$ of the sequence $s\left(t\right)$ is used, which defines the mean velocity of the AUV along a given trajectory, i.e., in this model, the mean velocity is modeled randomly for each trajectory but remains constant within an individual trajectory. The mean velocity for the entire ensemble of AUV trajectories is given by $E\left\{S(t)\right\}={\left(E\left\{s_{X}\left(0\right)\right\},E\left\{s_{Y}\left(0\right)\right\},E\left\{s_{Z}\left(0\right)\right\}\right)}^{\prime }$.

In the conducted experiment, the components of $s\left(0\right)$ are independent and uniformly distributed: $s_{X}\left(0\right)\sim R[-20,-10]$, $s_{Y}\left(0\right)\sim R[-20,-10]$, $s_{Z}\left(0\right)\sim R[-2,\;0]$. Thus, the average velocity of motion of A is characterized by the expected value $E\left\{S(t)\right\}=E\left\{s(0)\right\}={\left(-15,-15,-1\right)}^{\prime }$. From this, the absolute value of the mean velocity is ≈21 km/h, and it is directed toward $O\left(0,0,0\right)$. The covariance is $diag\left\{D\left\{s_{X}\right\};D\left\{s_{Y}\right\};D\left\{s_{Z}\right\}\right\}\approx diag\left\{2.9^{2},2.9^{2},0.4^{2}\right\}$. The standard deviations of the velocity disturbance vector $W_{t}={\left(w_{X}\left(t\right),w_{Y}(t),w_{Z}(t)\right)}^{\prime }$ are $\sigma_{s_{X}}=15,\sigma_{s_{Y}}=15,\sigma_{s_{Z}}=1\;km/h$. The velocity covariance is $cov\left(S(t),S(t)\right)\approx diag\left\{15.3^{2},15.3^{2},1.1^{2}\right\}$.

Note that the disturbance magnitudes are chosen such that $\sigma_{s_{X}}=\left|E\left\{s_{X}(0)\right\}\right|,$ $\sigma_{s_{Y}}=\left|E\left\{s_{Y}(0)\right\}\right|,\sigma_{s_{Z}}=\left|E\left\{s_{Z}(0)\right\}\right|.$ That is, the “magnitude” of the disturbance is approximately equal to the velocity value. The sole rationale behind this choice is to make the AUV motion highly chaotic at high speeds in order to create extremely challenging conditions for conducting model-based testing of the filtering algorithm.

It is easy to see that the moment characteristics of the velocity $S\left(t\right)$ are constant, namely$$\begin{array}{c} E\left\{S(t)\right\}={\left(E\left\{s_{X}\left(0\right)\right\},E\left\{s_{Y}\left(0\right)\right\},E\left\{s_{Z}\left(0\right)\right\}\right)}^{\prime },\\ \begin{array}{c} cov\left(S(t),S(t)\right)=diag\left\{D\left\{s_{X}\left(0\right)\right\}+\sigma_{s_{X}}^{2},D\left\{s_{Y}\left(0\right)\right\}+\sigma_{s_{Y}}^{2},D\left\{s_{Z}\left(0\right)\right\}+\sigma_{s_{Z}}^{2}\right\}=\\ \begin{array}{c} =diag\left\{D\left\{s_{X}\left(0\right)\right\}+{\left(E\left\{s_{X}\left(0\right)\right\}\right)}^{2},D\left\{s_{Y}\left(0\right)\right\}+{\left(E\left\{s_{Y}\left(0\right)\right\}\right)}^{2},D\left\{s_{Z}\left(0\right)\right\}+{\left(E\left\{s_{Z}\left(0\right)\right\}\right)}^{2}\right\}=\\ \begin{array}{c} =diag\left\{E\left\{s_{X}^{2}\left(0\right)\right\},E\left\{s_{Y}^{2}\left(0\right)\right\},E\left\{s_{Z}^{2}\left(0\right)\right\}\right\}. \end{array} \end{array} \end{array} \end{array}$$

As previously stated, we will not use model (16) but will instead improve upon it. First, we take into account the dependence of the “magnitude” of the disturbance on the velocity. This can be done following the example of a well-known aerodynamic effect, in which air resistance to a moving object increases with the square of its velocity. The same effect, naturally, occurs during motion in water. Of course, this is a rather complex physical phenomenon, the study of which requires considering drag and lift forces [48]. However, in our simple model of the motion of a material point, it is sufficient to assume that the disturbance variance depends on the current velocity. This approach allows, for instance, a more accurate representation of the influence of ocean currents on the motion of the AUV [5], which are not accounted for in existing models at all.

The “magnitude” of the disturbance will now be modeled not by the constant values $\sigma_{s_{X}},\sigma_{s_{Y}},\sigma_{s_{Z}}$, but by functions $\Sigma_{s_{X}}\left(S\right)$, $\Sigma_{s_{Y}}\left(S\right)$, $\Sigma_{s_{Z}}\left(S\right)$, $S=\left(S_{X},S_{Y}{,S}_{Z}\right)$. These functions continue to serve as standard deviations, i.e., they will act as scaling factors for the additive disturbances $w_{X}(t)$, $w_{Y}(t)$, $w_{Z}(t)$, which have unit variance.

Model (16) then takes the following new form:(17)$$\begin{array}{c} \begin{array}{c} X\left(t\right)=X\left(t-1\right)+\delta S_{X}\left(t\right),\;\;\;S_{X}\left(t\right)=s_{X}(0)+\Sigma_{s_{X}}\left(S\left(t-1\right)\right)w_{X}\left(t\right),\\ Y\left(t\right)=Y\left(t-1\right)+\delta S_{Y}\left(t\right),\;\;\;S_{Y}\left(t\right)=s_{Y}(0)+\Sigma_{s_{Y}}\left(S\left(t-1\right)\right)w_{Y}\left(t\right), \end{array}\\ Z\left(t\right)=Z\left(t-1\right)+\delta S_{Z}\left(t\right),\;\;\;S_{Z}\left(t\right)=s_{Z}(0)+\Sigma_{s_{Z}}\left(S\left(t-1\right)\right)w_{Z}\left(t\right). \end{array}$$

To select the forms of the functions $\Sigma_{s_{X}}(S),\Sigma_{s_{Y}}(S),$ and $\Sigma_{s_{Z}}(S)$, various assumptions may be employed. We define the deviation as a fraction (portion) of the absolute value of the velocity in the corresponding direction. Let these fractions be denoted by the constants $\varepsilon_{s_{X}},\varepsilon_{s_{Y}},$ and $\varepsilon_{s_{Z}}$. Then, we define as follows: $\Sigma_{s_{X}}\left(S\right)=\varepsilon_{s_{X}}\left|S_{X}\right|$, $\Sigma_{s_{Y}}\left(S\right)=\varepsilon_{s_{Y}}\left|S_{Y}\right|$, $\Sigma_{s_{Z}}\left(S\right)=\varepsilon_{s_{Z}}\left|S_{z}\right|$. It is easy to see that the projections $S_{X}\left(t\right)$, $S_{Y}\left(t\right)$, $S_{Z}\left(t\right)$ of the velocity of the AUV in model (17), defined in this way, may exhibit approximately the same moments as the velocity projections in the previous model. Specifically, the conditions $E\left\{S_{X}\left(t\right)\right\}=E\left\{s_{X}(0)\right\}$, $E\left\{S_{Y}\left(t\right)\right\}=E\left\{s_{Y}(0)\right\}$, $E\left\{S_{Z}\left(t\right)\right\}=E\left\{s_{Z}(0)\right\}$ are satisfied. The expression for the variance (considering $D\left[S_{x}\left(t\right)\right]$ as an example), assuming independence of $W_{t}$, is as follows:$$\begin{array}{c} D\left[S_{X}\left(t\right)\right]=D\left[s_{X}(0)\right]+\varepsilon_{s_{X}}^{2}E\left\{{\left|S_{X}\left(t-1\right)\right|}^{2}\right\}=\\ \begin{array}{c} =D\left[s_{X}(0)\right]+\varepsilon_{s_{X}}^{2}{\left(E\left[s_{X}(0)\right]\right)}^{2}+\varepsilon_{s_{X}}^{2}D\left[S_{X}\left(t-1\right)\right]. \end{array} \end{array}$$

This is the geometric progression sum. For $0<\varepsilon_{s_{X}}<1$, it converges to $\frac{E\left\{s_{X}^{2}(0)\right\}}{1-\varepsilon_{s_{X}}^{2}}-{\left(E\left[s_{X}(0)\right]\right)}^{2}.$ If we choose $\varepsilon_{s_{X}}^{2}=\frac{{\left(E\left[s_{X}(0)\right]\right)}^{2}}{{\left(E\left[s_{X}(0)\right]\right)}^{2}+E\left\{s_{X}^{2}(0)\right\}}$, then for sufficiently large t, we obtain $D\left[S_{X}\left(t\right)\right]\approx const=E\left\{s_{X}^{2}(0)\right\}$, as previously. A similar approach applies to the projections $S_{Y}\left(t\right)$ and $S_{Z}\left(t\right)$. We now have an explanation of how to select $\varepsilon_{s_{X}},\varepsilon_{s_{Y}},$ and $\varepsilon_{s_{Z}}$. For the above-specified distribution of s(0), we obtain $\varepsilon_{X}^{2}=\varepsilon_{Y}^{2}\approx 0.491$ and $\varepsilon_{Z}^{2}\approx 0.463$. In the calculations, the value $\varepsilon_{s_{X}}=\varepsilon_{s_{Y}}=\varepsilon_{s_{Z}}=0.7$ was used.

The second refinement of model (16) or (17) is the simulation of the possibility of a sudden change in the mean velocity. To this end, we employ a standard Poisson process P(u), independent of the position of A. Let $\lambda_{u}$ denote the intensity of changes in the constant mean velocity, and $\frac{1}{\lambda_{u}}$ the mean time between such velocity shifts. Discrete time t is related to continuous time u via the discretization step: u=tδ.

We use the jumps of P(tδ) and the sections of the previously introduced sequence s(t) to alter the mean velocity: the constant component of the velocity vector $S\left(t\right)$ (i.e., the first term) is updated at the moments of Poisson process jumps. To achieve this, we define the jump indicator $p\left(t\right)=\lambda_{t\delta }\left(\;P\left(t\delta \right)-P\left(\left(t-1\right)\delta \right)\right)$ and transform model (17) into the following form:(18)$$\begin{array}{c} \begin{array}{c} \begin{array}{c} X\left(t\right)=X\left(t-1\right)+\delta S_{X}\left(t\right),\;\;\;S_{X}\left(t\right)=s_{X}(t)+\Sigma_{s_{X}}\left(S\left(t-1\right)\right)w_{X}\left(t\right),\\ s_{X}\left(t\right)=\left(1-p\left(t\right)\right)s_{X}\left(t-1\right)+p\left(t\right)\left(s_{X}^{p}\left(t\right)-X\left(t-1\right)\right), \end{array}\\ \begin{array}{c} Y\left(t\right)=Y\left(t-1\right)+\delta S_{Y}\left(t\right),\;\;\;S_{Y}\left(t\right)=s_{Y}(t)+\Sigma_{s_{Y}}\left(S\left(t-1\right)\right)w_{Y}\left(t\right),\\ s_{Y}\left(t\right)=\left(1-p\left(t\right)\right)s_{Y}\left(t-1\right)+p\left(t\right)\left(s_{Y}^{p}\left(t\right)-Y\left(t-1\right)\right), \end{array} \end{array}\\ \begin{array}{c} Z\left(t\right)=Z\left(t-1\right)+\delta S_{Z}\left(t\right),\;\;\;S_{Z}\left(t\right)=s_{Z}\left(t\right)+\Sigma_{s_{Z}}\left(S\left(t-1\right)\right)w_{Z}\left(t\right),\\ s_{Z}\left(t\right)=\left(1-p\left(t\right)\right)s_{Z}\left(t-1\right)+p\left(t\right)\left(s_{Z}^{p}\left(t\right)-Z\left(t-1\right)\right), \end{array} \end{array}$$

Here, the independent variable $s_{X}^{p}\left(t\right)$ (and similarly $s_{Y}^{p}\left(t\right)$ and $s_{Z}^{p}\left(t\right)$) has the following distribution: $s_{X}^{p}(t)\sim R\left[a_{X},b_{X}\right]-\frac{a_{X}+b_{X}}{2}$, where $\left[a_{X},b_{X}\right]$ denotes the interval over which the initial mean velocity $s_{X}\left(0\right)$ is uniformly distributed (similarly for $\left[a_{Y},b_{Y}\right]$ and $\left[a_{Z},b_{Z}\right]$).

The proposed model has a very straightforward interpretation. The jumps of the Poisson process $\lambda_{u}P(u)$ determine the time moments when the mean velocity along the trajectory changes. The distribution of the “new” mean velocity remains uniform with the same average direction of motion from the A current position to the object O. The moments are preserved in the sense that $E\left\{\left.s_{X}(t)\right|p\left(t\right)=1,X\left(t-1\right)\right\}=-X\left(t-1\right)$ and $D\left\{s_{X}^{p}\left(t\right)\right\}=D\left\{s_{X}(0)\right\}$ (similarly for $s_{Y}(t)$ and $s_{Z}(t)$).

In the numerical experiment, the intensity $\lambda_{u}=\frac{3}{6}=0.5\;per\;minute$, i.e., on average, there are three changes in the mean velocity during the observation period, or the average time between jumps is 2 min. In all other respects, model (18) retains the same parameters as model (17).

### 3.2. Model of Stationary Observers

Observations of the AUV are carried out by two complexes, (F,first) and (S,second). The coordinate system Oxyz is oriented as follows. As shown in Figure 1, the z-axis corresponds to the depth of the AUV. In addition, the y-axis is directed from the first observer toward the object (F→O), and the x-axis is directed from the second observer (S→O). Thus, the coordinates of the observers are $F\left(X_{F},Y_{F},Z_{F}\right)=\left(0,Y_{F},0\right)$ and $S\left(X_{S},Y_{S},Z_{S}\right)=\left(X_{S},0,0\right)$. For the calculations, we used the values $X_{S}=-2\;km$ and $Y_{F}=-1\;km$. The AUV motion model described in Section 3.1 ensures that, throughout the entire observation period, the coordinates $A\left(X(t),Y(t),Z(t)\right)$ satisfy the following conditions: (1) the AUV remains at depth and does not surface, i.e., Z(t)>0; (2) for both observers, the condition $X_{A}>X_{M}$ is used in the formation of the pseudo-measurement (6) holds, i.e., X(t)>0. The experiment is schematically illustrated in Figure 2.

The position $A\left(X\left(t\right),Y\left(t\right),Z(t)\right)$ is modeled at discrete time moments t=0,…,1000; the observation time interval is discretized with a step of δ=0.0001h(h). Thus, tracking is performed over 0.1 h=6 min. With a constant absolute average speed of 21 km/h, during this time the AUV on average travels a distance of approximately 2.1 km, approaching O. The maximum distance from A to O and to F or S is approximately 28 km and 30 km, respectively. The minimum distances are approximately 144 km and 16 km. Accordingly, the maximum time delay possible at the moment of AUV detection is assumed to be T=56, i.e., 0.0056 h or approximately 20s(s), given the speed of sound in water $v_{s}\approx 5400\;km/h$ (1500 m/s).

Unlike model (14), the experiment does not include a “preliminary” interval t=−T−1,−T,…,0. Instead, in the calculations for t=0,…,56, the EKF estimate is not computed using Equation (15). Over this interval, instead of a filtering estimate, a simple geometric estimate is computed; to this end, noise in the received measurements is ignored, angles and range are converted into Cartesian coordinates for each beacon, and the two resulting estimates are averaged with equal weights.

The time delay Equation (13) in our example takes the following form:$$\begin{array}{c} \tau_{t}^{\left(1\right)}=\tau_{t}^{\left(F\right)}=min\left\{T,\left[\sqrt{{\left(X\left(t\right)\right)}^{2}+{\left(Y\left(t\right)-Y_{F}\right)}^{2}+{\left(Z\left(t\right)\right)}^{2}}/\left({\delta v}_{s}\right)\right]\right\},\\ \begin{array}{c} \tau_{t}^{\left(2\right)}=\tau_{t}^{\left(S\right)}=min\left\{T,\left[\sqrt{{\left(X\left(t\right)-X_{S}\right)}^{2}+{\left(Y\left(t\right)\right)}^{2}+{\left(Z\left(t\right)\right)}^{2}}/\left({\delta v}_{s}\right)\right]\right\}. \end{array} \end{array}$$

The observation vector in model (14) is given by $y_{t}={\left({\left(y_{t}^{\left(1\right)}\right)}^{\prime },{\left(y_{t}^{\left(2\right)}\right)}^{\prime }\right)}^{\prime }={\left(y_{\varphi_{t}}^{F},y_{\lambda_{t}}^{F},y_{r_{t}}^{F},y_{\varphi_{t}}^{S},y_{\lambda_{t}}^{S},y_{r_{t}}^{S}\right)}^{\prime }$, where

Bearings:


$$\varphi_{t}^{F}=arctan\frac{Y\left(t-{\tau_{F}}_{t}\right)-Y_{F}}{X\left(t{-\tau_{F}}_{t}\right)},\varphi_{t}^{S}=arctan\frac{Y\left(t-{\tau_{S}}_{t}\right)}{X\left(t{-\tau_{S}}_{t}\right)-X_{S}},$$


Elevation angles:


$$\lambda_{t}^{F}=arctan\left(\frac{Z\left(t-{\tau_{F}}_{t}\right)}{X\left(t-{\tau_{F}}_{t}\right)}cos\left(\varphi_{t}^{F}\right)\right),\lambda_{t}^{S}=arctan\left(\frac{Z\left(t-{\tau_{S}}_{t}\right)}{X\left(t-{\tau_{S}}_{t}\right)-X_{S}}cos\left(\varphi_{t}^{S}\right)\right),$$


Ranges:


$$r_{t}^{F}=\frac{Z\left(t-{\tau_{F}}_{t}\right)}{sin\left(\lambda_{t}^{F}\right)},r_{t}^{S}=\frac{Z\left(t-{\tau_{S}}_{t}\right)}{sin\left(\lambda_{t}^{S}\right)}.$$


Finally, the measurement accuracy parameters $y_{t}$:$$\begin{array}{c} cov\left(v_{t}^{F},v_{t}^{F}\right)=cov\left(v_{t}^{S},v_{t}^{S}\right)=diag\left\{\sigma_{\varphi },\sigma_{\lambda },\sigma_{r}\right\},\\ \begin{array}{c} \begin{array}{c} \sigma_{\varphi }=\sigma_{\lambda }=\frac{\pi }{180}\;rad\left(magnitude\;of\;the\;error\;1^{{}^{\circ}}\right),\\ \sigma_{r}=0.1\;km\;(magnitude\;of\;the\;error\;100\;m). \end{array} \end{array} \end{array}$$

And the final model refinement for the case of correlated measurement errors. We implement this idea for the range measurement $y_{r}$. Specifically, we assume that $\sigma_{r}=\sigma_{r}\left(r\right)$. That is, we proceed from the following considerations. Unlike angles, whose measurements always lie within the range [0,2π], range measurements vary differently. It is likely unreasonable to assume the same error for measuring the distance of a nearby object (hundreds of meters) and a distant object (several kilometers). It is more appropriate to express the error as a percentage, i.e., define $\sigma_{r}\left(r\right)=\varepsilon_{r}r$. This refinement does not require any changes in the previously discussed pseudo-measurements. Formally, we must specify that in model (14)$$\psi_{t}^{\left(i,2\right)}\left(X_{t}\right)=diag\left\{1,1,\varepsilon_{r}^{i}r\right\},\Psi_{t}^{\left(i,2\right)}=diag\left\{1,1,1,1,\varepsilon_{r}^{i}r\right\}.$$

Accordingly, the measurement and pseudo-measurement errors have$$cov\left(v_{t}^{\left(i\right)},v_{t}^{\left(i\right)}\right)=diag\left\{\sigma_{\varphi },\sigma_{\lambda },1\right\},\;cov\left(V_{t}^{\left(i\right)},V_{t}^{\left(i\right)}\right)=diag\left\{\sigma_{\varphi },\sigma_{\varphi },\sigma_{\lambda },\sigma_{\lambda },1\right\}.$$

In the calculations performed, we used the values $\varepsilon_{r}^{F}=\varepsilon_{r}^{S}=0,005$. Thus, at a distance of 20 km, the sensors will provide an error on the order of 0.1 km.

### 3.3. Numerical Experiments

It should be emphasized that the filter variants applicable to the described model are limited to the method of linear pseudo-measurements presented in this paper. The possible use of other suboptimal methods requires substantial modification, while the Bayesian filter [11] cannot be implemented due to excessive computational cost. The EKF in its standard form, without pseudo-measurements, can be applied without modification. However, even under simpler conditions, this filter proves to be divergent [7].

Using computer simulation of N=100,000 motion trajectories (18), and observations $y_{t}^{\left(F\right)}={\left(y_{\varphi_{t}}^{\left(F\right)},y_{\lambda_{t}}^{\left(F\right)},y_{r_{t}}^{\left(F\right)}\right)}^{\prime }$ and $y_{t}^{\left(S\right)}={\left(y_{\varphi_{t}}^{\left(S\right)},y_{\lambda_{t}}^{\left(S\right)},y_{r_{t}}^{\left(S\right)}\right)}^{\prime }$, estimates of the AUV position ${\hat{X}}_{t}={\left(\hat{X}\left(t\right),\hat{Y}\left(t\right),\hat{Z}\left(t\right)\right)}^{\prime }$ were computed according to formulas (15) for observation scenarios with time delays (T=56) and without (T=0). The estimation accuracy is determined by the root mean square deviations $\sigma_{\hat{X}}\left(t\right),\;\sigma_{\hat{Y}}\left(t\right),\;\sigma_{\hat{Z}}\left(t\right)$ (shown in the figures in meters), calculated by averaging the estimation errors over the simulated ensemble.

Figure 3 illustrates the experiment, with several examples of characteristic trajectories of the AUV position in coordinates $X\left(t\right),Y(t)$ and the corresponding estimates $\hat{X}\left(t\right),\hat{Y}\left(t\right)$. The depth (coordinate $Z\left(t\right)$) varies significantly less chaotically. Additionally, it should be noted in the figures that, among all simulated trajectories, the time delays varied from 55 to 25, which amounts to 9–20 s.

Note that, in Figure 3, the beginning of the motion is accompanied by a cluster of inaccurate estimates. This corresponds to the first 56 steps, during which a simple geometric estimate was computed. The root mean square deviations of this estimate are denoted by $\Sigma_{\hat{X}}\left(t\right),\;\Sigma_{\hat{Y}}\left(t\right),\;\Sigma_{\hat{Z}}\left(t\right)$. An illustration of the estimation accuracy is provided in Figure 4.

Other variants of the computations were performed: for the model without time delays (T=0) and for the model without velocity parameter jumps ($\lambda_{u}=0$). A formal comparison across all models is presented in Table 1. To characterize the accuracy, the root mean square deviations of the estimation errors were averaged over the trajectories; for example, for $\hat{X}\left(t\right)$, the quantities were computed as ${\hat{\sigma }}_{\hat{X}}=\frac{1}{1000}\sum_{t=1}^{1000}\sigma_{\hat{X}}\left(t\right)$ and ${\hat{\Sigma }}_{\hat{X}}=\frac{1}{1000}\sum_{t=1}^{1000}\Sigma_{\hat{X}}\left(t\right)$, and so on.

It should be noted that models with abrupt changes in the velocity parameter (the second and fourth rows) affect the result in fundamentally different ways depending on the presence or absence of temporal delays. When T=0, a change in the constant average velocity in model (18) improves the accuracy of position estimation. Apparently, this occurs due to the more predictable direction of the average velocity when its value changes as the object is approached. In contrast, in the model with T=56, the accuracy decreases, since the temporal delay prevents the correct association of the moments of velocity change with the incoming observations. It should be emphasized that all calculations were carried out under the assumption that the velocity parameter $s\left(t\right)$ is known. In a more realistic model, this assumption should be abandoned, and model identification should be carried out in parallel, which makes the problem significantly more complex.

## 4. Conclusions

### 4.1. Summary

The main conclusion drawn from the results of the conducted experiment is the justification of the ability of the presented linear pseudo-measurements filter to solve the state filtering problem in a model with time delays. Initially, we confirmed that the filter successfully addresses the estimation problem in a standard Markov model without delays. This was expected, as the pseudo-measurements method has demonstrated strong performance, and the presented alternative approach to angular measurement approximation generally follows the same conceptual framework. The model with time delays, however, is a fundamentally different matter. As shown in other studies, even highly stable filters may fail to handle this model, despite exhibiting excellent performance in the absence of delays [9].

It is also important that the numerical results are in good agreement with earlier computations [7,8,9,10], where other filtering methods were applied to similar models [15,16]. Attention should be paid to the assumption regarding the availability of information about a single parameter of the motion model—the constant average velocity, including during its abrupt changes. This assumption ensures, for example, that the simple geometric estimate provides nearly the same accuracy in the models with T=0 and T=56. We understand that such an assumption is unrealistic in practice, and the parameters of the motion model require identification. The possibility of identification and the parameter estimation algorithm is a separate and rather complex issue. This question has been thoroughly examined in other works [9,10], where the feasibility of obtaining good estimates under observation delays was demonstrated. The question of whether the parameters of the motion model can be identified using the pseudo-measurements filter remains open and should be addressed in future research [49].

### 4.2. Discussion

Nevertheless, there are other issues related to the pseudo-measurements filter as a suboptimal filtering method—issues typical of such heuristic algorithms. Despite the demonstrated effectiveness of the linear pseudo-measurements methodology, the pseudo-measurements filter retains its weakest feature, namely, a tendency to diverge. This effect manifested under the following modifications of the observation system model:rejection of the assumption of a known constant average velocity parameter and its replacement in the filter by the mathematical expectation (in which case the motion model becomes nonlinear);omission of the “preliminary” observation stage with the computation of a simple geometric estimate (as is typical in the pseudo-measurements filter, the initial estimates are set to the expected value of the initial position);movement of the trajectory closer to the coordinate planes, i.e., deterioration of the conditions for angular approximation (due to increased disturbance intensity or extended observation time).

It is intuitively clear that all the above situations should lead to a decrease in estimation accuracy, but a reasonably acceptable result is still expected. In reality, however, the computations exhibit trajectories along which the pseudo-measurements filter diverges, demonstrating the overall instability of the procedure. This leads to the conclusion that, while the linear pseudo-measurements methodology is highly effective, it should ideally be applied not only within the pseudo-measurements filter but also within other, more robust and stable filtering schemes, such as those proposed in methods [15,16].

What is also important and promising for this topic is the following. First, the method of linear pseudo-measurements is not the only successful suboptimal filter. It is quite possible that other methods, such as the particle filter or the sigma-point (unscented) filter, will also ensure success in the presented model with random time delays. They cannot be applied directly under the described observation conditions, but their adaptation is possible and may be of interest for further study. The second promising direction is the consideration of various prior uncertainties characteristic of practical applications. Indeed, so far, we have considered models in which all parameters have a unique probabilistic description. Even the complication concerning the identification of an unknown constant average speed [9,10] leaves the model fully determined. In practice, this is not the case, since certain factors describing the operation of the observation system are always a priori uncertain. Formulations with various options of prior uncertainty, as well as methods to address them, such as robust, minimax, and/or adaptive estimation, are well-known and therefore can be implemented in the formulation considered here. In the same direction, studies can be conducted on possible complex observation conditions, for example, failures of observation channels and the AUV reaching the boundaries of the observation area, accompanied by singularities in the pseudo-measurements.

## Figures and Tables

**Figure 1 sensors-25-03757-f001:**
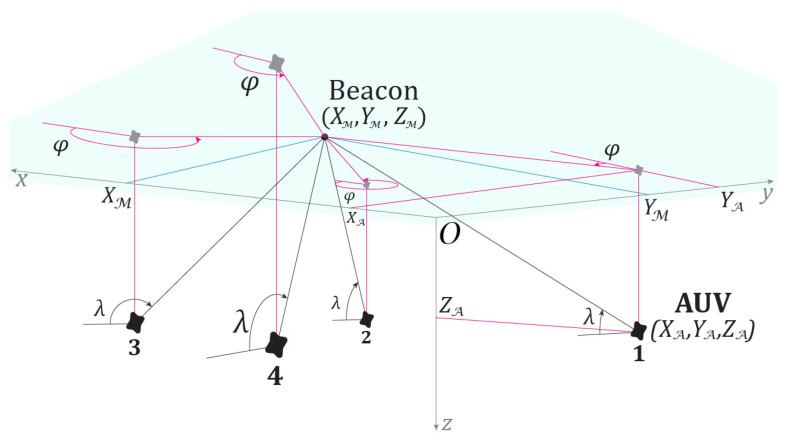
Mutual position of the AUV and the observer (underwater view): The AUV $\left(X_{A},Y_{A},Z_{A}\right)$ relative to each observer Beacon $\left(X_{M},Y_{M},Z_{M}\right)$ can be located in one of four positions: **1**—$X_{A}<X_{M}$, $Y_{A}>Y_{M}$; **2**—$X_{A}<X_{M}$, $Y_{A}<Y_{M}$; **3**—$X_{A}>X_{M}$, $Y_{A}<Y_{M}$; **4**—$X_{A}>X_{M}$, $Y_{A}>Y_{M}$.

**Figure 2 sensors-25-03757-f002:**
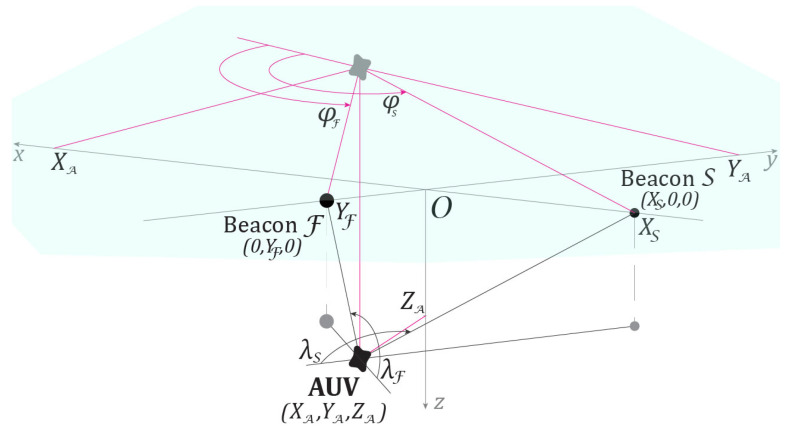
Observers and AUV arrangement scheme in the experiment (underwater view): The AUV in the initial position $\left(X_{A},Y_{A},Z_{A}\right)$ and two observers—the first Beacon$\left(0,Y_{F},0\right)$ and the second Beacon$\left(X_{S},0,0\right)$—are arranged in the experiment according to variant 4 in Figure 1.

**Figure 3 sensors-25-03757-f003:**
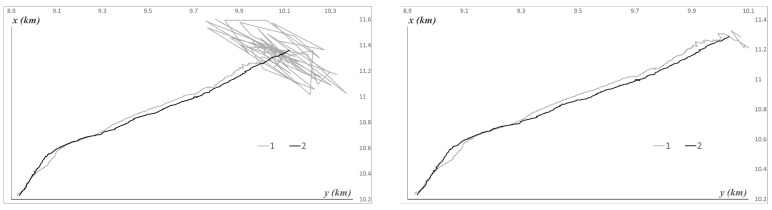
Examples of characteristic AUV trajectories and position estimates. 1—$\left(X\left(t\right),Y\left(t\right)\right)$, 2—$\left(\hat{X}\left(t\right),\hat{Y}\left(t\right)\right)$, on the left t=0,…,1000, on the right t=57,…,1000.

**Figure 4 sensors-25-03757-f004:**
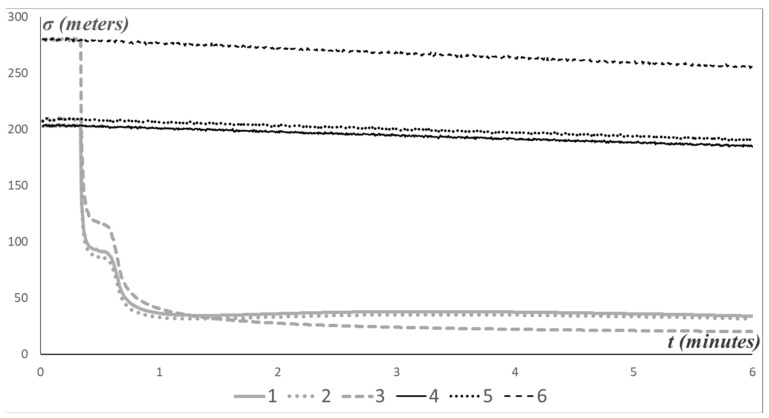
Root mean square deviations: 1—$\sigma_{\hat{X}}\left(t\right),$ 2—$\sigma_{\hat{Y}}\left(t\right),$ 3—$\sigma_{\hat{Z}}\left(t\right),$ 4—$\Sigma_{\hat{X}}\left(t\right),$ 5—$\Sigma_{\hat{Y}}\left(t\right),$ 6—$\Sigma_{\hat{Z}}\left(t\right)$.

**Table 1 sensors-25-03757-t001:** Comparison of accuracy of the AUV coordinate estimates (in meters).

**Model**	${\hat{\sigma }}_{\hat{X}}$	${\hat{\sigma }}_{\hat{Y}}$	${\hat{\sigma }}_{\hat{Z}}$	${\hat{\Sigma }}_{\hat{X}}$	${\hat{\Sigma }}_{\hat{Y}}$	${\hat{\Sigma }}_{Z}$
T=0 $\lambda_{u}=0$	24.81	23.34	26.22	193.79	199.29	267.02
T=0 $\lambda_{u}=0.5$	23.79	22.33	24.76			
T=56 $\lambda_{u}=0$	39.50	38.04	45.47	194.72	200.20	268.05
T=56 $\lambda_{u}=0.5$	49.06	46.63	45.37			

## Data Availability

The original contributions presented in the study are included in the article, further inquiries can be directed to the author.

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
