# Peer review of "Linear Pseudo-Measurements Filtering for Tracking a Moving Underwater Target by Observations with Random Delays"

_sensors, 2025, doi:10.3390/s25123757_

Round 1

Reviewer 1 Report

Comments and Suggestions for Authors

Dear Author

Shortcomings in the attached file.

Author Response

Dear reviewer!

Thank you so much for your work. Please find my answers in the attached file.

Reviewer 2 Report

Comments and Suggestions for Authors

This paper addresses the challenge of tracking moving objects in stochastic observation systems where random time delays occur between observation acquisition and the actual state of the object. The context includes limited prior knowledge of measurement error distributions, which are specified only by their first two statistical moments. This a topic of primary interest for the special issue “Advancements and Applications of Cooperative Positioning, Planning and Control for Autonomous Vehicles.”

The introduction provides sufficient background it is well presented. The authors propose the adaptation of a linear pseudo-measurements filter to accommodate the complexities of the observation model. The contribution relies on the fact that the linear pseudo-measurement method is adapted for a model with random time delays and under conditions of incomplete information about the error distribution of measurements, which is a challenging task.

The proposed method is applied to a scenario involving an autonomous underwater vehicle (AUV) navigating toward a stationary target. The vehicle's velocity is subject to continuous random disturbances and periodic abrupt changes, with observations conducted using two stationary acoustic beacons. The method effectively manages the uncertainties and maintains accurate tracking performance.

I only have a couple of recommendations to improve the paper:

  1. The methodology would benefit from a diagram that allows the user to follow some steps to implement the proposed adapted pseudo-measurements filter.
  2. The conclusions section can be separated into a discussion section (that must be expanded) and the conclusions. The discussion section should include the statements that demonstrate advances with respect to the state of the art, and the limitations (that where described).

Author Response

(The authors gave the same response as above.)

Reviewer 3 Report

Comments and Suggestions for Authors

This article approaches the azimuth-elevation range tracking problem through functional transformation of the angular measurements to replace the original non-linear motion model with a linear model. This allows the application of linear Kalman filtering to the transformed problem, subject to various small-angle approximations. The author then shows that the formalism can be extended to the case where the time-delays associated with measurements are not negligible, which is the case for tracking of underwater vehicles using active sonars, and when range sensing accuracy is proportional to range. Finally, the author applies the derived techniques to track a rapidly-moving underwater vehicle and demonstrates that tracking remains feasible even when the vehicle exhibits a highly unusual degree of variability in its motion.

This article is unusual, in that it is not an end-point. Rather, it is one line of inquiry among a number that the author has previously undertaken. The author notes that the method is robust and flexible, but its performance depends on unrealistic knowledge of a key parameter - the average speed of the vehicle - and that further work would be necessary to estimate that parameter from the data. The introduction suggests that a degree of predictability is possible using a pseudo-measurement approach that is not possible with other sub-optimal filters, but this is not entirely borne out by the analysis.

The article is extremely well-written and the English expression is word-perfect, but the text demands close attention from the reader. Multiple references must be consulted to fully understand the present work and figure captions are very terse. Figure 1 appears on page 3 and shows a number of symbols without explanation. It is not mentioned in the text before page 11, and some symbols are never explained. Later figures are somewhat easier to understand, but the random variation shown in Figure 3 during the geometric estimation phase of filtering somewhat obscures the convergence of the filter when it begins to act - convergence is only clearly visible in the lower-right example. 

In summary, while this work is a mathematical tour-de-force, its significance could perhaps be more clearly articulated, given that the author has been very clear about the limitations of the approach and the requirement for further work to achieve an approach with practical value.

• What is the main question addressed by the research?
The paper addresses the problem of tracking an object in 3 dimensions using a technique that linearizes angular measurements, thereby permitting application of a linear Kalman filter after various approximations. This permits the author to quantify the accuracy of the tracking filter to an extent, and also permits specific allowances for time delays arising from slow propagation speeds, and variations in the target being tracked. The caveat to this is that the model relies on fore-knowledge of the mean velocity of the target.

• Do you consider the topic original or relevant to the field? Does it address a specific gap in the field? Please also explain why this is/ is not the case. 
The paper seems to be quite original, and it fills a gap in the tracking literature, to which the author has previously contributed widely. The paper also discusses how the approach could be extended to make the approach more useful, by including velocity estimation.

• What does it add to the subject area compared with other published material?
The author is seeking to apply a new of approach to the field, seeking theoretical advantages in terms of predictability rather than a finished product.

• What specific improvements should the authors consider regarding the methodology? 
The figure captions are rather terse. Figure 1 shows a number of symbols that are not explained in the text, and Figure 1 is not referenced until page 11. The paper could be improved by explicitly explaining the details of the figure earlier in the text. Figure 3 shows that the track of the target has a large amount of variation before sufficient data is available to apply the filter. This somewhat obscures the convergence process. The figure could perhaps be improved by starting the filter output at the last moment before sufficient samples are available to apply the filter. Filter 4 could potentially be improved by adding limits to the plot showing reference values for uncertainty levels, although it is not clear how this could be done. The author notes that EKF solutions are divergent; therefore there is no value in including them.

• Are the conclusions consistent with the evidence and arguments presented and do they address the main question posed? Please also explain why this is/is not the case.
Yes. The author shows that the filter performs well against a target with artificially challenging behaviour.

• Are the references appropriate?
Yes, the references are appropriate and also very well explained.

Author Response

(The authors gave the same response as above.)

Round 2

Reviewer 1 Report

Comments and Suggestions for Authors

Dear Author,

thank you for making changes to the article I have no further comments.